# Identification of Viruses Infecting Oats in Korea by Metatranscriptomics

**DOI:** 10.3390/plants11030256

**Published:** 2022-01-19

**Authors:** Na-Kyeong Kim, Hyo-Jeong Lee, Sang-Min Kim, Rae-Dong Jeong

**Affiliations:** 1Department of Applied Biology, Institute of Environmentally Friendly Agriculture, Chonnam National University, Gwangju 61185, Korea; tails7724@naver.com (N.-K.K.); hjhjhj8@naver.com (H.-J.L.); 2Crop Foundation Research Division, National Institute of Crop Science, Rural Development Administration, Wanju-Gun 55365, Korea; kimsangmin@korea.kr

**Keywords:** barley yellow dwarf virus, cereal yellow dwarf virus, RNA-seq, oats

## Abstract

Controlling infectious plant viruses presents a constant challenge in agriculture. As a source of valuable nutrients for human health, the cultivation of oats (*Avena sativa* L.) has recently been increased in Korea. To date, however, few studies have been undertaken to identify the viruses infecting oats in this country. In this study, we carried out RNA-sequencing followed by bioinformatics analyses to understand the virosphere in six different geographical locations in Korea where oats are cultivated. We identified three different virus species, namely, barley yellow dwarf virus (BYDV) (BYDV-PAV and BYDV-PAS), cereal yellow dwarf virus (CYDV) (CYDV-RPS and CYDV-RPV), and rice black-streaked dwarf virus (RBSDV). Based on the number of virus-associated reads and contigs, BYDV-PAV was a dominant virus infecting winter oats in Korea. Interestingly, RBSDV was identified in only a single region, and this is the first report of this virus infecting oats in Korea. Single nucleotide polymorphisms analyses indicated that most BYDV, CYDV, and RBSDV isolates show considerable genetic variations. Phylogenetic analyses indicated that BYDVs and CYDVs were largely grouped in isolates from Asia and USA, whereas RBSDV was genetically similar to isolates from China. Overall, the findings of this study provide a preliminary characterization of the types of plant viruses infecting oats in six geographical regions of Korea.

## 1. Introduction

Plant viruses cause significant yield reduction and economic losses in many crops worldwide. Such viruses can be devastating, especially for varieties with no virus resistance [1,2]. Thus, early detection and identification of plant viruses are of considerable importance for controlling disease development and improving crop security.

Oat (*Avena sativa*), of the family Poaceae, is generally cultivated as a secondary crop. Oat was derived from a weed of the primary cereal and is ranked 7th in terms of cereal production after maize, wheat, barley, and sorghum [3,4]. However, although other cereal crops, such as wheat, rice, and barley, are generally of higher quality and are more profitable than oats for growers [5], oats contain larger amounts of valuable proteins and fat than most grains, in addition to providing an excellent source of fiber, which are considered effective in preventing heart disease [6,7]. Given these associated health benefits, the production and consumption of oats as a “super food” are gradually increasing in Korea.

In common with other crop plants, the production of oats can be severely affected by the invasion of emerging plant pathogens (bacteria, fungi, and viruses). Of identified viruses infecting oat, barley yellow dwarf virus (BYDV), cereal yellow dwarf virus (CYDV), oat blue dwarf virus (OBDV), and oat mosaic virus (OMV) have been widely documented worldwide. The most frequently identified viruses, BYDV (genus *Luteovirus*) and CYDV (genus *Polerovirus*) are single-stranded RNA viruses in the family *Luteoviridae,* which can detrimentally affect more than 150 species of the Poaceae family [8,9]. These viruses are transmitted via the phloem by various species of aphid vectors, and, indeed, the different serotypes of BYDV can be classified based on the transmitting aphids [10]. Presently, there are five species of viruses infecting cereals or grasses in the genus *Luteovirusa*: BYDV-PAV, BYDV-PAS, BYDV-MAV, BYDV-kerII, and BYDV-kerIII [11]. In a previous study, BYDV-PAS was distinguished from BYDV-PAV based on the nucleotide sequence of the CP gene [12,13,14]. BYDV-infected oats show symptoms such as stunted leaves, reduced root growth, and leaf discoloration (reddening), but infections may be asymptomatic [15,16]. BYDV has been reported to cause yield losses of up to 25% in barley, 46% in wheat, and 15% in oats, with average losses of 30% worldwide [17,18,19]. In the genus *Polerovirus*, five species of viruses infect cereals or grasses: CYDV-RPV, CYDV-RPS, maize yellow dwarf virus, maize yellow mosaic virus, and sugarcane yellow leaf virus [11]. Among these, CYDV (CYDV-RPV and CYDV-RPS) encode a P0 viral suppressor of RNA silencing that is not seen in BYDV [20], and these viruses, which can have severe effect on wheat growth, are the most commonly reported species [21]. 

Various detection assays such as polymerase chain reaction (PCR), quantitative real time polymerase chain reaction, enzyme-linked immunosorbent assay, and loop-mediated isothermal amplification can be conducted for specific viral diseases, given prior knowledge of the sequence or antibody. In contrast, next-generation sequencing (NGS) can detect pathogens without prior knowledge of the disease [22,23]. Thus, NGS technology has been applied both for known viruses and viroids in plant virology, and for novel viruses. NGS has been used in studies associated with virus populations, viral mutations, and evolution.

In the early 2000s, high-throughput sequencing (HTS)-based approaches using RNA-sequencing analysis were developed and used to identify viruses infecting different plant species [24,25,26,27,28]. HTS can also be employed to identify very low titers of viruses or viroids. Today, NGS technology for virus detection, which can identify many viruses in numerous crops, is emerging as the number of studies increase. HTS techniques can reveal both known and novel viral pathogens [29]. As previously reported, studies have identified unknown or novel viruses using RNA-sequencing for barley [30], apple [31], peach [32], sweet potato [27], garlic [33], and plum [34]. Each of these studies described analyses in agricultural crops and showed the advantages of HTS technology for identifying co-infection of pathogens. To date, however, there have been no comparable studies that have examined the viruses infecting oats in Korea. 

Several viruses infecting cereal crops have been previously studied using RNA-sequencing (RNA-seq) [11,30]. Although oat is an important cereal crop, the viruses that infect oats are still unclear. In this study, we carried out RNA-seq followed by our RNA-seq with BLASTN-based bioinformatics analyses to examine viruses infecting oats in Korea. Additionally, we revealed viral communities in six different geographical regions of oat fields in Korea. 

## 2. Results

### 2.1. Sample Collection and Library Generation for Identification of Viruses Infecting Oat Plants

To examine viruses infecting oat plants, we collected 323 oat samples from plantations located in six different regions of Korea from March to June 2020 (Table 1 and Figure 1a). We named each library according to the geographic region. All oats were grown in fields, as shown in Figure 1b. The collected oat samples showed orangey-brown blotches, reddening, and stunting (Figure 1c,d). We also collected leaf samples from various oat cultivars (Joyang, Samhan, and Taehan). Samples were pooled and subjected to total RNA extraction and library preparation from the six different regions for RNA-seq. Overall, we prepared six libraries, which were paired-end (2 × 100 bp) sequenced using the HiSeq4000 system.

### 2.2. Identification of Viruses from Oats’ Transcriptomes

The number of raw reads bases ranged from 5,504,951,432 (YP library) to 6,811,937,066 (JE library). The raw reads obtained by RNA-seq from the six libraries were individually de novo assembled using the gsAssembler program. The number of assembled contigs ranged from 365,264 (library IG) to 618,180 (library SW). These libraries were deposited in an SRA database with accession numbers (Table 2).

To examine virus-associated reads and contigs, the raw reads and assembled contigs were used for a BLASTN search against a virus reference genome database obtained from the NCBI database. The number of virus-associated reads ranged from 107,106 (HN) to 618,395 (YP) (Figure 2A). Additionally, we obtained a total of 5523 virus-associated contigs from the six libraries. The number of identified virus-associated contigs ranged from a minimum of seven (HN) to a maximum of 3113 (Figure 2B).

The virus-associated contigs from the dataset were first combined and then classified. We identified three different viruses, BYDV, CYDV, and RBSDV in all libraries (Figure 2C). Of these, BYDV-PAV was the most frequently identified virus based on the number of identified contigs in all libraries. BYDV-PAV (446 contigs) and RBSDV (123 contigs) were identified from YP. BYDV-PAV (3101 contigs) and CYDV-RPS (12 contigs) were identified from SW, whereas BYDV-PAV (59 contigs and seven contigs) was only identified from IG and HN, respectively. BYDV-PAS (453 contigs), BYDV-PAV (415 contigs), CYDV-RPS (91 contigs), and CYDV-RPV (83 contigs) were identified from JE, whereas BYDV-PAV (369 contigs), BYDV-PAS (352 contigs), and CYDV-RPS (2 contigs) were identified from GJ. Interestingly, RBSDV was identified only in YP.

### 2.3. Viral Genome Assembly and Analyses of SNPs of Identified Viruses

Through de novo assembly, we obtained virus consensus sequences mapped to each reference virus genome from the six libraries (Figure 3). Virus-associated contigs were mapped to near completion on the entire region of the virus genome without gaps. BYDV-PAV identified from library YP showed 94.2%, 96.6% (SW), 94.4% (IG), 99.2% (JE), 99.2% (HN), and 99.2% [35] identity (Figure 3) [35]. BYDV-PAS identified from JE and GJ mapped to 92.6% and 95.8%, respectively (Figure 4A). CYDV-RPS identified from SW, JE, and GJ showed 98.9%, 98.6% and 99.0% identity, respectively (Figure 4B). Additionally, CYDV-RPV derived from JE mapped to 97.9% (Figure 4C). RBSDV, only identified from YP, ranged from 70.1% (segment 2) to 99.3% (segment 3) (Figure 5). Six BYDV-PAV isolates, two BYDV-PAS isolates, three CYDV-RPS isolates, one CYDV-RPV isolate, and one RBSDV isolate, composed of 10 genomic segments from NGS analysis, were deposited in GenBank (Table 3).

The number of identified SNPs among each identified virus genome was diverse; these were analyzed by reads mapping to the virus reference genome (Appendix A). In the case of BYDV-PAV isolates from the six libraries, the number of SNPs ranged from 210 (SW) to 611 (JE). SNPs for BYDV-PASs (983, JE; 883, GJ) were higher than those for BYDV-PAVs (Figure 6A). Two CYDV isolates from JE showed 552 (CYDV-RPS) and 677 (CYDV-RPV) SNPs. For CYDV isolates derived from SW and GJ, 34 and 89 SNPs were identified, respectively (Figure 6B). Further, the number of SNPs of RBSDV that were composed of 10 segments identified from library YP ranged from 12 (segment 9) to 47 (segment 7) (Figure 6C).

### 2.4. Phylogenetic Analyses of Identified Viruses

We performed phylogenetic analyses with assembled viral genomes for BYDV isolates, CYDV isolates, and RBSDV isolates identified using RNA-seq. For phylogenetic analyses, we retrieved CP sequences of 16 BYDV isolates, 14 CYDV isolates, and 14 RBSDV isolates from the NCBI database. After nucleotide sequence alignment, three phylogenetic trees were constructed using MEGA 7 (Figure 7).

A phylogenetic analysis of the CP sequences of the BYDV isolated from all six libraries, in addition to other isolates from various countries, showed two separated clades. In the blue colored clade, the identified isolates (isolate PAV-YCPV; LC590229, isolate PAS-KJ; LC592173, isolate PAV-KJ; LC589963, isolate PAV-HN; LC589961, isolate PAV-IS; LC590228) obtained from RNA-seq were grouped into the same clade; they were closely related to BYDV-PAV isolates from USA (EF521840) and Australia (MK962883, X07653). In the orange clade, BYDV isolate (isolate PAV-SW; LC589965) was closely related to the China isolate (AY855920) (Figure 7A).

For CYDV, the phylogenetic tree of CYDV showed two defined groups, and four CYDV isolates identified from RNA-seq were separated into two clades (Figure 7B). Two CYDV-RPS isolates (LC589964 and LC589966) from SW and GJ were grouped into the same clade, and they were closely related to each other. CYDV-RPS (LC589967) and CYDV-RPV (LC590227) isolates from JE were grouped into the same clade, and they were closely related to each other. For RBSDV, the phylogenetic tree generated using CPs of known RBSDV isolates showed that most RBSDV isolates were grouped into two clades. RBSDV-YCPC isolate from YP was closely related to China isolates in the same clade (Figure 7C).

### 2.5. Validation of the Identified Viruses Using RT-PCR

To confirm the presence of viruses identified by RNA-seq, we carried out RT-PCR using specific primers previously used for the detection of these viruses (Table 4). The RT-PCR results were similar to the RNA-seq analysis results (Figure 8). BYDV-PAV and -PAS infections in the six libraries were confirmed using RT-PCR. The infection of CYDVs was confirmed in the SW, JE, and GJ libraries by RT-PCR using specific primers, which detected CYDV species. CYDV-RPV and -RPS showed quite similar genome sequences; thus, no specific primers sets are available to differentiate them. Additionally, RT-PCR also confirmed the presence of RBSDV in only YP library.

## 3. Discussion

We are becoming increasingly familiar with the concept that a variety of organisms, including plants, co-exist with a rich diversity of microorganisms—the microbiome. However, a somewhat less-considered notion is that organisms also interact with a considerable diversity of viruses, designated the virome, which can be defined as virus-associated nucleic acids existing anywhere in plants, animals, or the environment. Recently, the generic detection and analyses of highly diverse viruses in a range of sample types have been performed using HTS as a platform offering an ideal and unique approach [40,41]. The ongoing reduction in the cost of HTS, along with improvement in the tools used for HTS data analysis, are increasingly enabling researchers to address long-standing challenges, such as the identification of unknown viruses, the complexity of virus genomes, viral mutation, and virus replication, thereby providing unprecedented insights into plant viromes in a given environment [42,43,44]. However, although the viromes of a range of crop plants, including horticultural species, have already been the subject of intensive research [22,25,30,32,34], there have been comparatively few studies on the viromes of cereal crops [30,45], despite their considerable economic importance. Given their rich nutrient contents, oats are increasingly being cultivated in Korea, and it is accordingly deemed necessary to gain at least a preliminary understanding of the spectrum of viruses infecting oats nationwide [46]. To the best of our knowledge, there have to date been no surveys of viral diseases affecting oats in Korea, and thus in the present study, we sought to address this deficiency by conducting a survey of the viruses infecting oats in six different Korean provinces based on RNA-seq. RNA-seq easily identifies both RNA and DNA viruses by incorporating de novo assembly because viruses containing the DNA genome transcribe mRNA during virus replication [47]. Recently, several viruses infecting cereal crops, including maize yellow mosaic virus and wheat leaf yellowing-associated virus, have been successfully identified using RNA-seq [48,49]. 

In the present study, we pooled all samples collected from sites in the six surveyed provinces to monitor virus infections in a large number of samples. The pooling of samples not only reduces HTS costs and analysis time, but also provides broad-scale information on the structure of viral communities affecting oats in different geographical locations [30]. Our RNA-seq with bioinformatics analyses of field samples from six different regions in Korea revealed that different types of BYDVs (PAV and PAS), CYDVs (RPS and RPV), and RBSDVs were identified in different provinces. These results indicate that this was the first report of BYDV, CYDV, and RBSDV infecting oats in Korea. The assembled virus-associated contigs were nearly completed assembly corresponding target virus genomes, including BYDV-PAV, BYDV-PAS, CYDV-RPS, CYDV-RPV, and RBSDV. Similar previous survey studies conducted in Korea for different cereal crops and virome studies on barley have identified barley yellow mosaic virus, barley mild mosaic virus, hordeum vulgare endoavirus, barley virus G, and BYDV species PAV, PAS, and MAV belonging to the genus *Luteovirus* [30,50]. Given that the present study is the first to identify BYDV CYDV, and RBSDV viruses in Korean oat samples based on HTS analyses, further research is needed to evaluate the risks posed by domestic oat-infecting viruses using pest risk analysis, a risk assessment tool used in conjunction with international phytosanitary standards (ISPM 2 and ISPM 11) [51].

BYDV and CYDV are the most economically important viruses infecting cereal crops worldwide. These viruses differ in terms of ease of transmission by different aphid species. Among the viruses identified to infect oats, BYDV and CYDV are the most common viruses transmitted by aphids. For example, it has been reported that BYDV-PAV is transmitted by *Rhopalosiphum padi*, *Sitobion avenae*, *Metopolophium dirhodum* and *Sitobion fragariae*; BYDV-PAS by *Rhopalosiphum maidis*; CYDV-RPV by *Rhopalosiphum padi*, *Schizaphis graminum*; and CYDV-RPS by *Rhopalosiphum padi* [10]. Among these viruses, BYDV-PAV has been identified as being the most common and, compared with BYDV-PAS, is notably invariant across diverse geographical locations [52,53,54]. Unlike most BYDVs, CYDV isolates showed considerable variation. Whereas CYDV-RPS isolates appear to be common in North America, CYDV-RPV is believed to predominate in Europe [11]. RBSDVs, which comprise 10 double-stranded RNA segments, can be found in cereal crops in China, Japan, and Korea, and have been identified as a source of serious disease outbreaks causing substantial yield losses of rice and maize crops in China [55,56]. RBSDV propagates in vivo in their hopper vectors in a persistent manner [57]. In this study, BYDV-PAV and BYDV-PAS were identified in all six libraries. Of these, BYDV-PAV was identified as the predominant virus, accounting for more than 50% of viruses in all six libraries, based on the read and copy numbers associated with identified viruses. BYDV-PAS infection was detected at only two locations, in JE and GJ. Our findings thus appear to contrast with those reported for a survey study of BYDV infections of cereals crops in the Czech Republic, which revealed BYDV-PAS to be dominant compared with other BYDV species [12]. Such differences in the profiles of BYDV species affecting cereal crops in different countries have been suggested to be associated with a diverse range of factors, namely, the cultivation of different cereal crop cultivars, alternative hosts, different environmental conditions, and the diversity of aphid species transmitting these viruses. Additionally, CYDV-RPS and CYDV-RPV were identified in SW, JE, and GJ. These findings would thus tend to indicate that BYDVs are predominant virus for cultivated oats in Korea; however, we could not confirm whether BYDV directly affects virus infection symptoms and production losses, nor how BYDV has become a dominant virus in the evolution and ecology of oats. Consequently, further studies are necessary to investigate virus diversity from the perspectives of viral mutualism and antagonism and the association between BYDV and insect vectors or hosts. Economic losses due to viruses infecting oats can range from 13–25 kg/ha for every 1% increase in incidence, according to previous studies [15]. To date, however, there have been no comparable studies conducted in Korea that have assessed the productivity of domestic oats relative to the incidence of infection, and thus further epidemiological studies are needed. Although we detected a comparatively low diversity among the viral species infecting oats, their effects on oat production may be influenced to varying extents by plant virus–vector interactions in response to the ongoing changes in global climate. It is widely predicted that climate change will modify the characteristic of plant virus infection, resulting in an escalation of epidemic outbreaks. Moreover, such changes will almost certainly alter vector population numbers, growth rates, fecundity, and feeding. Accordingly, to ensure the sustainable production of oats, considerably greater emphasis should be placed on the need for extensive research on the impact of climate change on virus infection and productivity.

Viruses can evolve rapidly by mutation and recombination during replication, and thus, it can be expected that large-scale genome rearrangements and SNP mutations will occur naturally among virus isolates or strains. To confirm such variability in our Korean isolates, we examined SNPs for the eight assembled BYDVs, four assembled CYDVs, and ten segments of RBSDV genomes. The SNP analyses suggested that most BYDVs, CYDVs, and RBSDVs showed strong genetic variations. Among these, BYDV-PAS (983 SNPs) and CYDV-RPV (677) identified from JE exhibited the highest number of SNPs. An increased genetic variation in one region compared to very low genetic variation in another region can be thought of as a pooled oat sample for library construction. For example, in the case of JE library, which pooled the largest number (128) of samples, it was found that the largest number of SNPs were identified. In addition, the library with the lowest number of SNPs in BYDV-PAV was the SW library that pooled the fewest samples (16). However, BYDV-PAV identified in the YP library pooled with a small number (16) of samples showed a high number of SNPs. This suggests that there is no overall correlation with the number of pooled samples. Virus mutation is closely related to host type, complexity of plant–virus–insect relationships, and environmental conditions; thus, the correlations between the SNPs and geographical regions should be further examined. With respect to RBSDV, we found that the number of SNPs identified among YP isolates ranged from 12 (segment 9) to 47 (segment 7), indicating that there were no strong genetic variations among the genome segments.

Isolates of BYDVs and CYDVs collected in different countries have been found to show diversity with respect to their genome sequences and pathogenicity [54], and in this regard, our phylogenetic analyses based on CP gene sequences revealed that the BYDV and CYDV isolates identified in the present study can be grouped in two different clades. These findings thus tend to highlight that different geographical regions contribute to the genetic diversity of these viruses. In the case of RBSDV, the nucleotide diversity of S10, which encodes the RBSDV coat protein, may be higher than that of other segments due to its self-interaction, and is also independent of geographic location in both rice and maize [58]. Our phylogenetic analysis based on CP sequences indicated that RBSDV-YCPC isolates cluster in a clade along with isolates from China, regardless of host or geographical location.

In view of the genetic diversity and variability among the genomes of B/CYDV species, it has been suggested that the primers for each species are extremely important criteria. In this regard, a detection method for BYDV species has previously been developed based on the use of multiplex RT-PCR [57]. Accordingly, although in the present study we confirmed the presence of BYDV-PAV and BYDV-PAS in oats, based on RT-PCR analysis using previously reported primers, given the limited differences between the CYDV-RPV and CYDV-RPS genomes, it remains a challenge to differentiate these species using RT-PCR. Thus, we resorted to using universal primers in our RT-PCR analyses, which can detect these viruses only to the CYDV level.

In summary, in this study, we identified viruses infecting oats in different geographical regions of Korea. Our findings will contribute to gaining a more comprehensive understanding of the viral communities infecting this crop and thereby provide information that could be used to develop management strategies for the control of viral diseases in oats cultivated in Korea. However, given the limited scope of this study, we did not seek to estimate the direct crop losses attributable to these viruses nationwide. Given the current and predicted changes in global climate and the increasing diversity of insect vectors, HTS-based studies of the viromes of cultivated crops and alternative hosts as virus reservoirs are urgently needed to assess the prevalence and potential impact of viruses and to provide further insights on their epidemiological status.

## 4. Materials and Methods

### 4.1. Sample Preparation

From March to May 2020, oat leaves were collected from six different geographical regions in Korea. A total of 322 samples were collected from various cultivars (Joyang, Samhan, Hi-early, and Taehan). The symptomatic oat leaves showed reddening and dwarfing characteristics. Samples were frozen in liquid nitrogen and stored at −80 °C in a deep freezer until used. Samples were pooled based on geographical province for total RNA extraction and library preparation.

### 4.2. RNA Extraction and Library Preparation for RNA Sequencing (RNA-Seq)

Total RNA was extracted from the ground pooling samples using the Clear-S Total RNA Extraction Kit (InVirusTech Co., Gwangju, Korea), according to the manufacturer’s instructions. The quality and quantity of extracted total RNA were determined using an Agilent 2100 Bioanalyzer (Agilent Technologies, Palo Alto, CA, USA). The confirmed total RNA was used in the preparation of a cDNA library after ribosomal RNA was removed using a Ribo-zero rRNA removal kit (Illumina, San Diego, CA, USA) and purified and collected using RNA Clean XP. Purified mRNAs were fragmented and used for the synthesis of cDNA. Briefly, mRNAs with poly-A tail were extracted, and the first strand of cDNA was synthesized using the purified mRNAs followed by the second strand of cDNA. Then, the adenylation of 3′ ends was conducted using A-tailing mix, followed by washing using AMPure XP beads. To enrich DNA fragments that were ligated with adaptors, PCR was conducted. The quality and quantity of the generated libraries were determined using an Agilent D5000 ScreenTape system (Agilent Technologies, Palo Alto, CA, USA). Finally, oat leaves collected from Yeoncheon and Pocheon were named YP, Suwon as SW, Iksan and Gimje as IG, Jeongeup as JE, Haenam as HN, and Gangjin as GJ. The six libraries were loaded into flow cells, forming a library cluster. Then, five flow cell libraries were paired-end sequenced using a HiSeq 4000 sequencer, generating 200 bp paired-end reads (Macrogen, Co., Seoul, Korea). All raw sequences were deposited in the SRA database in NCBI.

### 4.3. Transcriptome Assembly and Virus Identification

A workstation (two six-core CPUs and 256 GB RAM) with the Ubuntu 12.04.5 LTS operating system was used for the de novo transcriptome assembly and BLAST search. Raw sequence reads obtained from each library were subjected to de novo transcriptome assembly using gsAssembler v2.8 (Roche Diagnostics, Branford, CT, USA) with default parameters [28]. To identify virus-associated contigs in the libraries, the assembled transcriptome contigs from each library were blasted against sequences in the viral reference database in NCBI (downloaded from https://www.ncbi.nlm.nih.gov/genome/viruses/ accessed on 16 April 2021) using BLASTX with a cutoff E-value of 1 × 10^−5^ [59]. The obtained virus-associated contigs were again subjected to a BLASTn search with an E-value cut off <1 × 10^−10^ against the mitochondria, chloroplast, and other contaminated oat host sequences of less than 200 bp derived from the NCBI database. Finally, only virus-associated contigs were selected for further virome analyses.

### 4.4. Viral Sequence Mapping and Genome Assembly

The sequences of all virus-associated contigs were compared with the identified virus reference genomes using BLAST. Contigs with the highest similarity matches to specific viruses were selected. The virus-associated contigs were mapped to identified reference virus genome sequences using Geneious v. 11.1.5 [60], and consensus sequences for the mapping file were generated based on an identity threshold of 95%.

### 4.5. Phylogenetic Analyses of Identified Viruses

For phylogenetic tree analyses, we generated complete coat protein (CP) gene sequences for BYDV, CYDV, and rice black-streaked dwarf virus (RBSDV) based on mapping consensus sequences. Phylogenetic trees for each virus were constructed with other viral CP sequences downloaded from GenBank. NCBI accession numbers of all viral genomes are indicated in the phylogenetic trees. Nucleotide sequences were aligned using CLUSTALW in BioEdit version 7.0.5.3., and the aligned nucleotide sequences were used to construct phylogenetic trees using 1000 bootstrap replicates based on either the neighbor-joining or the Kimura 2-parameter method in MEGA 7 [61].

### 4.6. Identification of SNPs for Assembled Virus Genome

Single nucleotide polymorphisms (SNPs) were identified by mapping reads sequence data to the consensus sequences for each complete or nearly complete viral genome, using Geneious v. 11.1.5 with default parameters, and then Geneious v. 11.1.5 was used to perform SNP-calling and to export an Excel file containing the SNP data [60]. The positions of identified SNPs on each viral reference mapping genome were visualized.

### 4.7. Confirmation of Identified Virus Presence by RT-PCR

To confirm the presence of identified viruses by RNA-seq in oat samples, the total RNAs were extracted from leaf pooling samples based on province using the Clear-S Total RNA Extraction Kit (InVirusTech Co., Gwangju, Korea). Three specific primer sets were designed to amplify genomic regions corresponding to the CP genes from BYDV, CYDV, and RBSDV. The RT-PCR was performed using the SuPrimeScript RT-PCR Premix (GeNet Bio, Daejeon, Korea), according to the manufacturer’s recommendations. Each of the 20 μL reaction mixtures included 0.4 μM forward primer, 0.4 μM reverse primer, 10 μL SuPrimeScript RT Premix, 2 μL RNA template, and 6 μL DEPC-treated water. The thermocycling conditions were as follows: a reverse transcription stage of 50 °C for 30 min; an initial denaturation step of 95 °C for 5 min; 35 cycles of 95 °C for 30 s, 56 °C for 60 s, and 72 °C for 60 s; and a final extension step of 72 °C for 10 min. After conducting RT-PCR, the amplification products were visualized by gel electrophoresis using 1.2% agarose gels under UV light. Furthermore, the amplicons were individually cloned into the pGEM-T Easy Vector (Promega, Madison, WI, USA), followed by Sanger sequencing.

### 4.8. Data Availability

The raw dataset generated in the present study will be available upon publication in the NCBI Sequence Read Archive (SRA) repository under accession numbers SRR13259170, SRR13259239, SRR13259172, SRR13259168, SRR13259171, and SRR13259240. The viral genome sequences obtained from this study were also deposited in GenBank with individual accession numbers.

## Figures and Tables

**Figure 1 plants-11-00256-f001:**
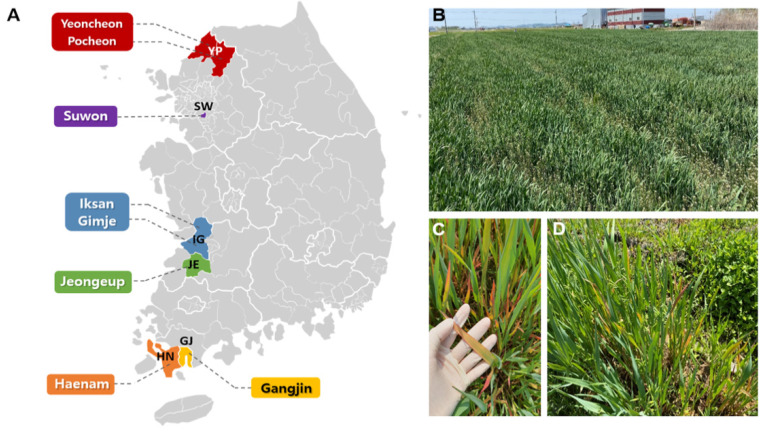
Locations of oats’ sample collection in Korea (**A**) and the viral symptoms of oat plants (**B**–**D**).

**Figure 2 plants-11-00256-f002:**
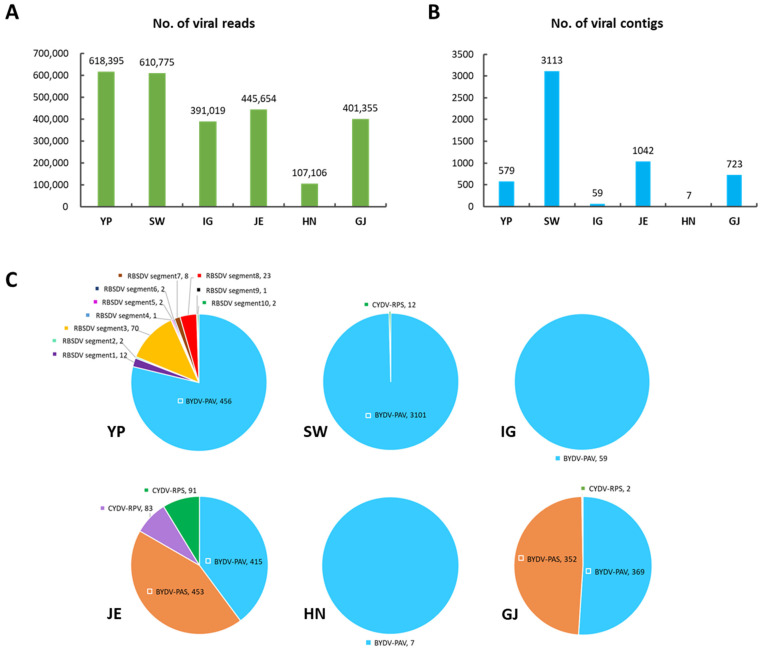
The number and proportion of virus-associated reads and contigs in each library. (**A**) The number of virus-associated reads in each library. (**B**) The number of virus-associated contigs in each library. (**C**) The proportion of identified viruses based on the number of virus-associated contigs in each library.

**Figure 3 plants-11-00256-f003:**
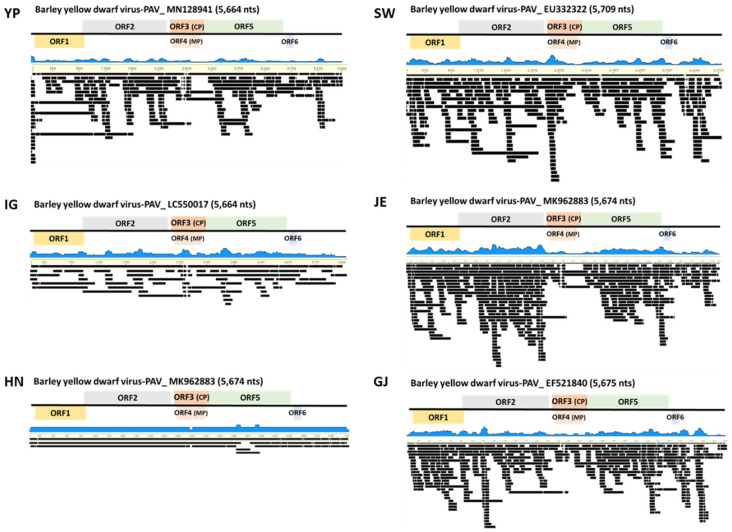
Viral genome assembly of identified BYDV-PAV isolates. Genome organization of assembled BYDV-PAV isolates genome of YP, SW, IG, JE, HN, and GJ. Viral genomes were obtained from assembled virus-associated contigs and their mapping on the reference virus genome.

**Figure 4 plants-11-00256-f004:**
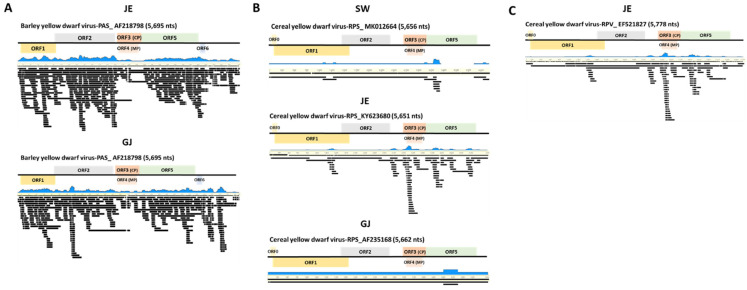
Genome assembly of identified BYDV-PAS isolates, CYDV-RPS, and CYDV-RPV isolates. Genome organization of assembled BYDV-PAS isolates genome from JE and GJ (**A**), CYDV-RPS isolates genome from SW, JE, and GJ (**B**), CYDV-RPV isolate genome from JE (**C**). Viral genomes were obtained from assembled virus-associated contigs and their mapping on the reference virus genome.

**Figure 5 plants-11-00256-f005:**
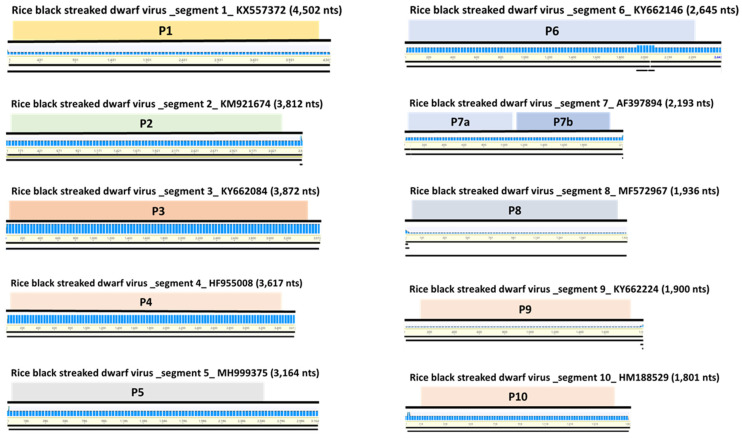
Genome assembly of identified RBSDV. Genome organization of each segment of identified RBSDV isolate from YP.

**Figure 6 plants-11-00256-f006:**
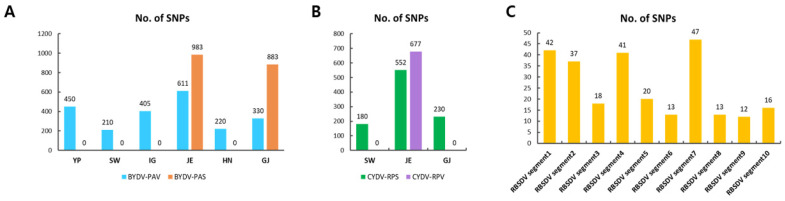
Identification of SNPs for identified viruses in each library. (**A**) The number of identified SNPs for BYDV-PAV and BYDV-PAS. (**B**) The number of identified SNPs for CYDV-RPS and CYDV-RPV. (**C**) The number of identified SNPs for RBSDV segments.

**Figure 7 plants-11-00256-f007:**
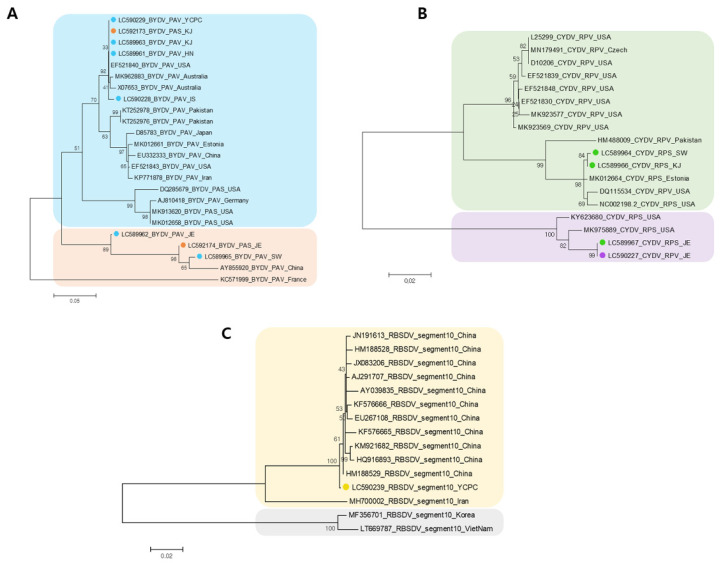
Phylogenetic trees for the identified BYDVs, CYDVs, and RBSDV. For the construction of the phylogenetic tree, coat protein (CP) nucleotide consensus sequences of identified viruses including BYDV (**A**), CYDV (**B**), and RBSDV (**C**), were assembled by RNA-seq. Assembled virus CP nucleotide sequences and other virus isolates CP nucleotide sequences obtained from GenBank were used for construction of phylogenetic trees. Phylogenetic trees were constructed using the MEGA7 program with maximum likelihood method and bootstrap with 1000 replicates. The identified virus isolates from this study were indicated by the different colors.

**Figure 8 plants-11-00256-f008:**
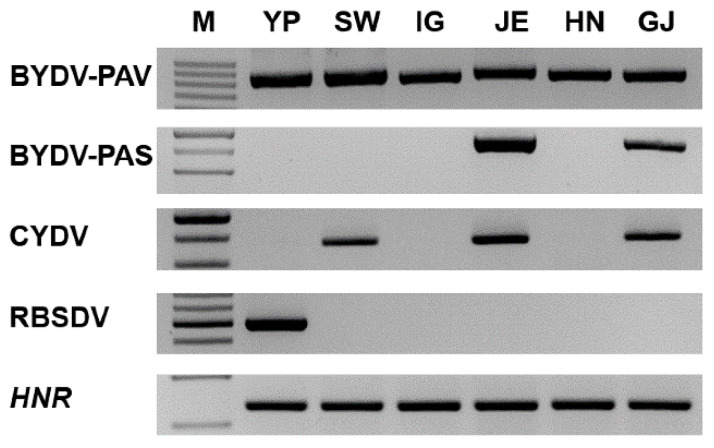
Confirmation of the identified viruses infecting oats by RT-PCR. RT-PCR results with specific primers; BYDV-PAV, BYDV-PAS, CYDV, and RBSDV. The *HNR* was used as an internal control [39]. Amplified PCR products were visualized by 1.2% agarose gel electrophoresis. The total RNA extracted for RNA-seq were used for RT-PCR.

**Table 1 plants-11-00256-t001:** Detailed information for name of library, sample location, and number of samples in each library.

Name of Library	Region	Location	Cultivar	No. of Samples
YP	Yeoncheon	38.1129° N, 127.0818° E	Joyang	12
	Pocheon	37.9422° N, 127.1930° E	Joyang	4
SW	Suwon	37.2741° N, 126.9884° E	mixed	16
IG	Iksan	35.9127° N, 126.9082° E	Samhan	21
JE	Jeongeup	35.6674° N, 126.9668° E	Joyang	49
		35.6499° N, 126.9356° E	Joyang	36
		35.6725° N, 126.8461° E	Joyang	43
HN	Haenam	34.6360° N, 126.6220° E	Joyang	22
GJ	Gangjin	34.6128° N, 126.7601° E	Joyang	119
Total				322

**Table 2 plants-11-00256-t002:** Summary of paired-end sequencing results for oat viromes using HiSeq4000 system.

Name of Library	Total Read Bases (bp)	Total Reads	GC (%)	Clean up Contig (e-Value ≤ 1 × 10^−5^)	SRA Accession Number
YP	5,504,951,432	36,456,632	49.47	600,485	SRR13259170
SW	6,227,005,346	41,238,446	47.27	618,180	SRR13259239
IG	5,981,435,254	39,612,154	45.47	365,264	SRR13259172
JE	6,811,937,066	45,112,166	46.18	585,956	SRR13259168
HN	6,156,541,498	40,771,798	49.46	503,397	SRR13259171
GJ	6,353,319,262	42,074,962	46.83	585,956	SRR13259240

**Table 3 plants-11-00256-t003:** List of identified viruses in six libraries in Korea.

Library	Virus	Strains/Segment	Isolate	Accession No.
YP	Barley yellow dwarf virus	PAV	BYDV-PAV-YCPV	LC590229
	Rice black streaked dwarf virus	segment 1	RBSDV-S1-YCPV	LC590230
		segment 2	RBSDV-S2-YCPV	LC590231
		segment 3	RBSDV-S3-YCPV	LC590232
		segment 4	RBSDV-S4-YCPV	LC590233
		segment 5	RBSDV-52-YCPV	LC590234
		segment 6	RBSDV-S6-YCPV	LC590235
		segment 7	RBSDV-S7-YCPV	LC590236
		segment 8	RBSDV-S8-YCPV	LC590237
		segment 9	RBSDV-S9-YCPV	LC590238
		segment 10	RBSDV-10-YCPV	LC590239
SW	Barley yellow dwarf virus	PAV	BYDV-PAV-SW	LC589965
	Cereal yellow dwarf virus	RPS	CYDV-RPS-SW	LC589964
IG	Barley yellow dwarf virus	PAV	BYDV-PAV-GJIS	LC590228
JE	Barley yellow dwarf virus	PAV	BYDV-PAV-JE2	LC589962
		PAS	BYDV-PAS-JE	LC592174
	Cereal yellow dwarf virus	RPS	CYDV-RPS-JE	LC589967
		RPV	CYDV-RPV-JE	LC590227
HN	Barley yellow dwarf virus	PAV	BYDV-PAV-HN2	LC589961
GJ	Barley yellow dwarf virus	PAV	BYDV-PAV-KJ2	LC589963
		PAS	BYDV-PAS-KJ	LC592173
	Cereal yellow dwarf virus	RPS	CYDV-RPS-KJ	LC589966

**Table 4 plants-11-00256-t004:** Information of primers used for RT-PCR.

Name of Virus	Primer Name	Sequence 5′-3′	Product Size (bp)	Reference
BYDV-PAV	ShuF	TACGGTAAGTGCCCAACTCC	831	[36]
	YanR	TGTTGAGGAGTCTACCTATTTG		
BYDV-PAS	Fwd	GGAGACGACTGTGTCATCATCACTGAG	448	[37]
	Rev	TGTCGTTTGTGATAGGTGTCTCC		
CYDV	Fwd	TCACCTTCGGGCCGTCTCTATCAG	372	[36]
	YanR	TGTTGAGGAGTCTACCTATTTG		
RBSDV	Fwd	TGGCTGTACCTTGTTTTGAT	501	[38]
	Rev	GACAATAGCTGAATTTCCCCC		
HNR ^1^	Fwd	ATTGGGTTTGTCACTTTCCGTAG	134	[39]
	Rev	CTTGGAGGGTGTCTCGCATCT		

^1^*Heterogeneous nuclear ribonucleoprotein 27C* (*HNR*); Internal control in oats (*Avena sativa* L.).

## Data Availability

The data presented in this study are available in the GenBank database (www.ncbi.nlm.nih.gov (accessed on 15 December 2021)) under accession numbers listed in the text.

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
