# Peer review of "Identification of Viruses Infecting Oats in Korea by Metatranscriptomics"

_plants, 2022, doi:10.3390/plants11030256_

Round 1
Reviewer 1 Report
The manuscript under review presents very interesting research results. Using the RNAseq transcriptome analysis, the authors identify viruses that cause oat viral diseases. Methodically, the experiment was properly planned. The sample selection is correct. They represent various areas of Korea where oats are grown, the importance of which is growing in this country due to the nutritional value of this cereal. The subject matter of the work is interesting, but since I have been working on Avena sativa for over 25 years, I cannot agree with some of the statements in the Introduction or Discussion. I marked them yellow in the attached file. In addition, the elements of the manuscript, which are general in nature and are intended to introduce the reader to the discussed issues, are written very poorly. I believe they should be improved. These are not only stylistic errors resulting from the lack of knowledge of the language but also a lack of factual information. As the authors note, there are few works devoted to oats, so I believe that this work has a chance to be read by many people interested in this cereal and should be improved. I have no objections to the methodology and materials. The results are very well presented, but the Abstract, Introduction and Discussion read very badly. I propose to correct these pieces of work. Add some substantive information and correct the English grammatically and stylistically. In summary, the revised manuscript after the corrections has a chance to be a highly cited publication, but it requires corrections, which I wrote about earlier.
Author Response
It is my great pleasure to resubmit a revised version of our manuscript (ID: plants-1534643) to plants. Many thanks for your decision letter and for allowing us to revise this manuscript. We have carefully considered the reviewer-1 comments and have incorporated all the suggested changes into this revised manuscript. We would like to thank the reviewer-1 for their critical evaluation of this manuscript. Changes in the revised manuscript are indicated by red color. Followings are the major changes included in the revised manuscript:
1. Line 22: it
As the reviewer 1 suggested, we have replaced it to its. Thank you.
2. Line 26: consortium a set
As the reviewer 1 suggested, we have revised consortium to a set. Thank you.
3. Line 39: Thus, there is a lack of agricultural research on oats worldwide.
The reviewer 1 mentioned that this sentence is not true. Thus, we have deleted this sentence.
4. Line 69: “with a complex of virus populations” what does it mean?
As the reviewer 1 suggested, we have revised it.
5. Line 94: presenting what does it mean?
We have revised “presenting” to “from”. Thank you.
6. Line 224: “The HNR was used as an internal control” add reference
As the reviewer 1 suggested, we have added a reference.
7. Line 276: in vivo italic
As the reviewer 1 suggested, we have revised it.
8. Lines 309: “China”
As the reviewer 1 suggested, we have revised “China” to “isolates from China”.
9. Line 327: “statuses”
As the reviewer 1 suggested, we have revised “statuses” to “status”
10. L342: “production”
As the reviewer 1 suggested, we have revised “production” to “preparation”
11. Line 364: https://www.ncbi.nlm.nih.gov/genome/viruses/ remove the link
As the reviewer 1 suggested, we have remove the link.

Reviewer 2 Report
The paper describes a very well performed, analyzed and presented study of viruses in oats in Korea applying RNA-Seq. The results are interesting, as different regions all across the country were examined and still a low diversity of viruses was identified, which is quite promising for the oat cultivation. I would expect some more input from the side of the authors regarding interpretation of results and their importance for the applied agriculture.
- l 45.: “and oat blue dwarf virus”, eliminate “and”
- 114: Please provide information about the length of the contigs, e.g. minimum, maximum and average length in bp.
- 141:” BYDV-PAV identified from library YP showed 94.2%, 96.6% (SW), 94.4% (IG), 99.2% (JE), 99.2% (HN), and 99.2% identity.” Please clarify against which reference sequence are these identities calculated.
- 146: “composed of ten segments from NGS analysis”. Do you mean 10 scaffolds? Or variants?
- 7, legend: Please add that CP sequences were used for the phylogenetic analysis. I also miss in the text the info whether nucleotide or amino acid sequences were used.
- 322: “this study identified viruses infecting oats in different geographical regions of Korea, contributing a comprehensive understanding of viral communities, that may be used to manage viral diseases in oats cultivation in Korea” . On that point I would like to underline, the following; From first detection of viral sequences through metagenomics till the point of taking management actions to control a viral disease the route is really long. This pathway is very well highlighted in the article by Massart et al. “A Framework for the Evaluation of Biosecurity, Commercial, Regulatory, and Scientific Impacts of Plant Viruses and Viroids Identified by NGS Technologies. Front. Microbiol. 8, 45. DOI: 10.3389/FMICB.2017.00045”, which I suggest to take into consideration.
- How do you evaluate your results? What does the low viral species diversity mean for the oat production? What does the increased genetic variation in only one region mean in comparison to the very low one in the other regions?. These questions arise and need to be discussed.
Author Response
It is my great pleasure to resubmit a revised version of our manuscript (ID: plants-1534643) to plants. Many thanks for your decision letter and for allowing us to revise this manuscript. We have carefully considered the reviewer-2 comments and have incorporated all the suggested changes into this revised manuscript. We would like to thank the reviewer-2 for their critical evaluation of this manuscript. Changes in the revised manuscript are indicated by red color. Followings are the major changes included in the revised manuscript:
Reviewer 2
1. “and oat blue dwarf virus”, eliminate “and”
As the reviewer 2 suggested, we have removed it. Thank you.
2. 114: Please provide information about the length of the contigs, e.g. minimum, maximum and average length in bp.
As the reviewer 2 suggested, we have added information about the length of the contigs. Thank you.
3. 141:” BYDV-PAV identified from library YP showed 94.2%, 96.6% (SW), 94.4% (IG), 99.2% (JE), 99.2% (HN), and 99.2% (GJ) identity.” Please clarify against which reference sequence are these identities calculated.
Thank you for your comments. We already suggested reference sequences in each library in Figure 3. For instance, BYDV-PAV contigs identified from YP assembled with MN128941.
4. 146: “composed of ten segments from NGS analysis”. Do you mean 10 scaffolds? Or variants?
It means that 10 genomic segments. RBSDV identified from oats in this study is composed of 10 genomic segments.
5. 7, legend: Please add that CP sequences were used for the phylogenetic analysis. I also miss in the text the info whether nucleotide or amino acid sequences were used.
We have added CP sequences for the phylogenetic analysis. Thank you.
6. 322: “this study identified viruses infecting oats in different geographical regions of Korea, contributing a comprehensive understanding of viral communities, that may be used to manage viral diseases in oats cultivation in Korea” . On that point I would like to underline, the following; From first detection of viral sequences through metagenomics till the point of taking management actions to control a viral disease the route is really long. This pathway is very well highlighted in the article by Massart et al. “A Framework for the Evaluation of Biosecurity, Commercial, Regulatory, and Scientific Impacts of Plant Viruses and Viroids Identified by NGS Technologies. Front. Microbiol. 8, 45. DOI: 10.3389/FMICB.2017.00045”, which I suggest to take into consideration.
Thank you for useful reference. We addressed shortly the process of newly identified viruses based on suggested reference. Thank you. Please see lines 270-273.
7. How do you evaluate your results?
Identification of virus using HTS in plants is useful approach. More importantly, validation is critical issue for presence of viruses in HTS-based approach. In this study, we confirmed the presence of identified viruses using RT-PCR and sanger sequencing. These validation assays were already mentioned in manuscript.
8. What does the low viral species diversity mean for the oat production?
We have addressed it. Please see lines 276-281 and 301-311. Thank you.
9. What does the increased genetic variation in only one region mean in comparison to the very low one in the other regions?
We have addressed it. Please see lines 317-325. Thank you.

Reviewer 3 Report
Oat is one of the important crops because of its benefit for human health. However, little attention was paid previously. This manuscript identified viruses infecting oats in Korea by RNA-seq and verified by RT-PCR. Although the methods are not novel, the manuscript contains new information and will be useful for controlling viral diseases in the future. It will be more interesting if the author could provide the data of viruses incidence in the six locations. The manuscript was designed well and presented in a logical way. The language of this manuscript needs further revision. I recommend accepting this manuscript.
Some minor revisions are needed:
Line 40, protein, revise to proteins;
Line 232, ribonucleoprotein , internal not international;
Reference 1, the name of the journal; Reference 49, F1000res,
Author Response
It is my great pleasure to resubmit a revised version of our manuscript (ID: plants-1534643) to plants. Many thanks for your decision letter and for allowing us to revise this manuscript. We have carefully considered the reviewer-3 comments and have incorporated all the suggested changes into this revised manuscript. We would like to thank the reviewer-1 for their critical evaluation of this manuscript. Changes in the revised manuscript are indicated by red color. Followings are the major changes included in the revised manuscript:
1. Line 40: proteins, revise to proteins
As the reviewer 3 suggested, we have revised it. Thank you.
2. Line 232: ribonucleoprotein, internal not international
As the reviewer 3 suggested, we have revised them. Thank you.
3. Reference : the name of the journal; reference 49, F1000res
As the reviewer 3 suggested, we have revised them. Thank you.

Round 2
Reviewer 1 Report
Language still needs revision.
Author Response
It is my great pleasure to resubmit a revised version of our manuscript (ID: plants-1534643) to plants. Many thanks for your decision letter and for allowing us to revise this manuscript. We have carefully considered the reviewer-1 comments and have incorporated all the suggested changes into this revised manuscript. We would like to thank the reviewer-1 for their critical evaluation of this manuscript. Changes in the revised manuscript are indicated by red color. Followings are the major changes included in the revised manuscript:
Reviewer 1
1. Language still needs revision.
As the reviewer 1 requested, we extensively edited English language in manuscript. Please see the manuscript.